# Production of Metallic Alloy Nanowires and Particles Templated Using Tomato Mosaic Virus (ToMV)

**DOI:** 10.3390/nano13192705

**Published:** 2023-10-05

**Authors:** Sachin N. Shah, Jonathan G. Heddle, David J. Evans, George P. Lomonossoff

**Affiliations:** 1Department of Biochemistry and Metabolism, John Innes Centre, Norwich Research Park, Norwich NR4 7UH, UK; 2Heddle Initiative Research Unit, RIKEN, 2-1 Hirosawa, Wako 351-0198, Saitama, Japan; jonathan.g.heddle@durham.ac.uk; 3Department of Chemistry, University of Hull, Hull HU6 7RX, UK; dave.evans@jic.ac.uk

**Keywords:** tomato mosaic virus, tobacco mosaic virus, nanowires, nanorods, nanoparticles, PVP, gold, platinum, palladium, XPS

## Abstract

We demonstrate a simple, low-energy method whereby tomato mosaic virus (ToMV) particles can be used to template the production of nanowires and particles consisting of alloys of gold (Au), platinum (Pt) and palladium (Pd) in various combinations. Selective nanowire growth within the inner channel of the particles was achieved using the polymeric capping agent polyvinylpyrrolidone (PVP_K30_) and the reducing agent ascorbic acid. The reaction conditions also resulted in the deposition of alloy nanoparticles on the external surface of the rods in addition to the nanowire structures within the internal cavity. The resulting materials were characterized using a variety of electron microscopic and spectroscopic techniques, which revealed both the structural and chemical composition of the alloys within the nanomaterials.

## 1. Introduction

The fabrication of metallic alloy nanoparticles (NPs) and nanowires (NWs) is a highly attractive approach to creating materials with enhanced optical, catalytic and photocatalytic properties [1]. For example, platinum is used as a catalyst for a broad range of electrochemical applications, including hydrogen evolution reactions (HER) [2,3] and oxygen reduction reactions (ORR) [4,5]. The incorporation of other metals, such as cobalt, palladium or gold, can enhance both HER and ORR [6,7,8]. The use of an AuPt alloy can also reduce the problem of poisoning of the platinum catalyst in electrochemical reactions [9]. Furthermore, AuPd NWs show much higher ORR than NWs made of individual metals due to electronic changes, strain and lattice distortion within the alloy [10]. However, it can be challenging to produce AuPtPd, AuPt and AuPd alloyed NPs of uniform dimensions using purely chemical approaches, with the production of NWs of consistent length being a particular issue [11,12,13,14,15,16]. Different metals combined with Pt to form alloys have varying kinetics of nucleation and growth. Co-reduction and thermal decomposition procedures may be insufficient to control the size and composition of trimetallic nanostructures [16].

As an alternative to direct chemical synthesis, highly ordered biological structures have been used to template the synthesis of nanomaterials. This approach has the advantages that milder conditions can be used and greater uniformity can be achieved. Examples of such biological templates include bacteria [17,18], fungi [19], algae [20], yeast [21,22] and virus particles [23,24]. Rod-shaped plant viruses, such as tobacco mosaic virus (TMV) and tomato mosaic virus (ToMV), have been used as templates for the synthesis of metallic nanomaterials. ToMV and TMV are closely related viruses in the genus Tobamovirus, which share 84% identical and 93% homologous amino acids within their coat proteins [25,26,27]. In each case, the particles are hollow cylinders measuring 300 nm in length with an outer diameter of 18 nm and a central channel diameter of 4 nm [28]. In the case of TMV, the electrostatic charge difference between the inner and outer capsid surfaces plays a key role in the selective deposition of metals such as Ni, Co and Cu [24,29,30,31,32]. These viruses have also been used for metal cluster deposition, the growth of NWs [33,34,35,36] and the production of nanotubes [37]. The deposition of the two-metal combinations CoFe, CoPt and FePt_3_ on either the interior or exterior surface of the rod-shaped viral particles has also been achieved using electroless deposition. In addition, ToMV has been used to template the deposition of iron oxide [27,38] on the outer surface of the particles and platinum on either the inner or the outer surface [37,38].

Here, we present the synthesis of metal alloy NWs bio-templated by the inner surface of ToMV particles, and deposition of alloy NPs on the outer surface. Two- or three-metal combinations of Au, Pt and Pd were deposited using a simple, wet-chemical metal reduction reaction using ascorbic acid (AA) at room temperature and the polymeric capping agent polyvinylpyrrolidone (PVP). This resulted in the production of alloy metal (M^0^) NWs measuring 4 nm in diameter and NPs, both of which had crystalline morphology.

## 2. Materials and Methods

### 2.1. Synthesis of Trimetallic ToMV NWs (AuPtPd ToMV NWs)

Tomato mosaic virus (ToMV) particles (a gift from Mime Kobayashi, NARA, Osaka, Japan) as a 1 mg/mL stock suspension in sodium phosphate buffer, pH 7.0, were dialyzed against Milli-Q water (resistivity 18.2 MΩ cm) for 12 h with two changes of water. The suspension was then diluted to 0.2 mg/mL.

A 200 µL sample of the diluted ToMV suspension was first incubated with an equal volume of freshly prepared aqueous 5 mM sodium tetrachloropalladate (II) (Na_2_PdCl_4_; Aldrich chemicals, Tokyo, Japan) for 20 min at 25 °C with 30 s of ultra-sonication (Branson 1510 DTH, 40 kHz, Tokyo, Japan) at 5 min intervals. The excess Na_2_PdCl_4_ was removed via centrifugation using a 100 kDa-molecular-weight cut-off Amicon centrifugal filter unit and washed twice with 100 μL Milli-Q water. The resulting concentrated, activated rods (20 µL final volume) were incubated with a freshly prepared premixed 1:1:1 solution of the three metal salt precursors Pt (potassium tetrachloroplatinate (II) (K_2_PtCl_4_, Aldrich chemicals, Japan)), Pd (sodium tetrachloropalladate (II) (Na_2_PdCl_4_; Aldrich chemicals, Japan)) and Au (tetrachloroauric (III) acid (HAuCl_4_; Aldrich chemicals, Japan)), each at a starting concentration of 1 mM, to give a final volume of 200 μL. The reaction mixture was vortexed for 60 s followed by 30 s of ultra-sonication, and then, left at room temperature for 5 min. The capping agent polyvinylpyrrolidone (PVP_K30_; Mw 40,000 g/mol, Nacalai tesque Kyoto, Japan, 5 μL of 23 mg/mL) was added, mixed well, and the reaction mixture sonicated for 30 s. A total of 10 μL of 10 mM freshly prepared ascorbic acid (Wako pure chemical industries Ltd., Osaka, Japan) was added and the reaction mixed well and sonicated for 5 s. The reaction was then subjected to 2 s cycles of sonication over a period of 30 s. After approximately 1 min, the reaction solution slowly became pale-yellow, which changed to a brownish-red over 30 min. To ensure the complete reduction of HAuCl_4_, K_2_PtCl_4_ and Na_2_PdCl_4_, the reaction was incubated overnight, becoming a darker brownish-red color.

### 2.2. Synthesis of Bimetallic ToMV NWs (AuPt, AuPd and PtPd ToMV NWs)

In parallel experiments, two metal precursors were used instead of three for the generation of bi-metallized ToMV particles without prior activation of the surface with palladium (Pd II). Various metal (Au, Pt and Pd) combinations were used for the metallization of ToMV particles under identical conditions to those used for the fabrication of tri metallic NWs within the 4 nm diameter channel.

### 2.3. Recovery of Bi- and Tri-Metallized ToMV Particles

The ToMV-metallized reaction samples were centrifuged at 4000 rpm for 3 min. The supernatant was decanted, and the pellet was resuspended in 200 μL Milli-Q water. This step was repeated to recover the purified metallized sample before further dilutions.

### 2.4. Electron Microscopy (EM)

A suspension of purified metallized nanorod (4 μL droplet) was placed on a carbon-coated copper (C-Cu) EM grid. Excess sample was removed using filter paper and the grid washed with Milli-Q water (20 μL). Excess water was removed from the C-Cu EM grid surface and the grid left to air dry for 5 to 10 min. The resulting grids were kept overnight in a sealable, vacuumed glass desiccator to absorb any remaining moisture. Sample grids were observed via transmission electron microscopy (TEM) using a JEOL JEM-1230 at 80 kV and a high-resolution transmission electron microscope (HR-TEM) using a JEOL JEM-2100F (Tokyo, Japan) at 200 kV. Scanning transmission electron microscopy (STEM), selected area electron diffraction (SAED) and energy-dispersive X-ray spectroscopy (EDX) were performed using a JED-2200 analyzer (JEOL, Tokyo, Japan). Elemental color mapping was conducted using a TEM model JEOL JEM-2100F (Tokyo, Japan) at 200 kV. The electron beam was focused to 1 nm and signals were integrated for 30 s. For the EDX, a barium specimen holder and C-Cu EM grids were used.

### 2.5. X-ray Photoelectron Spectroscopy (XPS) and Argon (Ar^+^) ion-Sputtered Sample Analysis

A thermally stable silicon oxide wafer was cut from a 100 mm × 0.50 mm diameter disk into small pieces. The P-type wafer consisted of a 300 nm SiO_2_ layer on Si (100), 1SP R: 0.01–0.05 Ωcm (MTI corporation, Richmond, CA, USA, SI-SO-Ba100D05C1-300 nm). The piece of silicon wafer (5 mm × 5 mm) was treated for 30 min in an ultra-sonicator (Branson 1510 DTH) in acetone, 2-propanol and water at 40 kHz. The wafer was cleaned in 150 mL water and 5 mL concentrated HCl and heated to 70 °C, followed by the slow addition of 5 mL 30% (*v/v*) H_2_O_2_. After 10 min, the wafer was thoroughly washed with Milli-Q water and dried in a nitrogen stream. Glow discharge hydrophilization was achieved via a 300 s treatment in a hydrophilic treatment device (JEOL, HDT-400, Tokyo, Japan).

The purified nanowire sample (1.5 µL droplet) was placed on a piece of 4 mm × 4 mm pre-treated silicon wafer and dried in air. The sample-loaded silicon wafer was kept overnight in a sealable, vacuumed glass desiccator to absorb any remaining moisture from the sample. XPS measurements were recorded using a VG ESCALAB 250 spectrophotometer (Thermo Fisher Scientific K. K. Tokyo, Japan), with monochromatic A1 Kα X-ray radiation (1486.6 eV), at room temperature. The instrument was operated at 200 W, the acceleration voltage set at 15 kV and the base chamber pressure kept at less than 10^−8^ Pa. Binding energies were calibrated against carbon as a reference (C 1s 284.1 eV). The effect of argon (Ar^+^) sputtering on the nanowire samples was studied. Argon sputtering was carried out using an EX-05 ion gun in the XPS analysis at a 1 µA beam current of 1 keV ions for 10 s. The measurements were performed using a low-energy electron flood gun. During the electron dosing, a charge neutralizer was used. An earthed conductive grid and carbon conductive tape were attached around the area.

## 3. Results and Discussion

We adapted an ambient wet synthesis technique to template alloy nanomaterial synthesis using ToMV particles under relatively mild chemical conditions. By manipulating various combinations of the Au, Pt and Pd salt precursors, we were able to correspondingly vary the constituent metals (Au, Pt and Pd) within the material; this allowed for the formation of alloys with a tunable composition. Three- (AuPtPd) and two-metal-combination (AuPt, AuPd and PtPd) alloys could therefore be selectively synthesized. Fabrication was carried out by incubating purified ToMV particles with different combinations of the desired metal salt precursors, HAuCl_4_, K_2_PtCl_4_ and Na_2_PdCl_4_, followed by reduction with ascorbic acid in the presence of the capping agent PVP_K30_ [38]. Ascorbic acid is an electron donor and PVP serves to control metal NP growth in the reaction process. Sonication was applied during the metallization reaction as this has been shown to increase the length and uniformity of the resulting nanomaterial [36,38]. Overall, the metallization reactionship can be described as follows:ToMV + M^2+^ + C_6_H_8_O_6_ → ToMV M^0^ NW + 2H^+^ + C_6_H_6_O_6_

### 3.1. Synthesis of ToMV Gold–Platinum–Palladium Nanomaterials (ToMV AuPtPd)

During the formation of the three-metal AuPtPd alloy, a change in color from pale-yellow to brownish-red was observed after the addition of ascorbic acid (Appendix A). After removing excess reactants, the sample was initially characterized via transmission electron microscopy. In the absence of staining, the sample appeared to consist of nanowires (NWs; dark lines) in excess of 100 nm in length, surrounded by approximately spherical NPs 15–20 nm in diameter, which appeared to have variable density (Figure 1a,b). In parallel control experiments, either the ToMV template or the capping agent was absent; in the first case, only larger-sized NPs were formed (Appendix A), whereas without a capping agent, there was no NW formation (Appendix A). Examination of the sample negatively stained with 2% (*w/v*) aurothioglucose, which does not penetrate the inner channel of ToMV [38] (Appendix A), revealed that the NWs lay within the central cavity of the viral particles, while the NPs decorated the external surface (Figure 1c,d).

To evaluate the elemental composition of the NWs and NPs, transmission electron microscopy–energy-dispersive X-ray spectroscopy (TEM-EDX) was performed on the unstained metalized sample. This revealed the presence of all three metals, Au, Pt and Pd, within the area analyzed (Figure 2a,b); the signals for copper (Cu) and carbon (C) were derived from the carbon-coated copper grid. High-angle annular bright-field scanning transmission electron microscopy (HAABF-STEM EDX) of a single metallized ToMV particle, consisting of both an NW and NPs, suggested that all three metals, Au, Pt and Pd, were distributed along its length (Appendix A). An examination of the area in Figure 2b at a higher resolution (Figure 2c and Appendix A) suggested that the NWs were 4 nm in diameter, corresponding to the diameter of the central channel of ToMV, and that the material in both NWs and NPs was crystalline. The measurement of the lattice fringes of the NW gave *d*-spacings of 2.28 Å or 1.97 Å (Figure 2c and Appendix A), depending on the area analyzed, while the analysis of NPs gave *d*-spacings of 2.27 Å and 1.98 Å for the denser and less dense NPs, respectively (Figure 2c).

These figures correspond to the interplanar distance assigned to the (111) and (200) planes for the metals involved. Fast Fourier transform (FFT) images generated using Image J software (Figure 2c inset) from the areas within the colored squares in Figure 2c confirmed the crystalline nature of the metals and the assigned facets.

To determine whether the metal content of an NW is consistent along its length, HAABF-STEM EDX was performed (Figure 2d). The resulting line profile showed that while all three metals were present, their relative abundance varied along the length of the NW. Similar analysis of the selected NPs suggested that the less dense NPs consist mainly of Pd while the denser NPs contain similar proportions of all three metals (Figure 2e). To obtain a more detailed view of the crystalline structure of NWs and NPs, a single metallized ToMV particle was imaged using high-angle annular scanning transmission electron microscopy under both dark- and bright-field conditions (HAADF and HAABF STEM; Figure 3a,b). An area from the bright-field image (Figure 3c) was subjected to selected area electron diffraction (SAED). The resultant diffraction pattern (Figure 3d) of concentric rings with bright spots was consistent with a face-centered cubic (*fcc*) metal crystal with lattice planes (111) and (200), and (220) and (311).

The binding energies (BE), chemical valency state and surface composition of tri-metallized ToMV particles were analyzed via X-ray photoelectron spectroscopy (XPS; Figure 4 and Appendix A). A typical separated doublet in the XPS was observed in the spectrum for each metal element, Au, Pt and Pd, within the particles. In each case, the doublets corresponding to the BEs were shifted to higher values compared to their bulk reference values [39]; Appendix A) surface cleaning and etching via argon ion (Ar^+^) sputtering of the sample for 10 s (green line) or 20 s (orange line) resulted in a shift in the BE to lower values compared to the same sample observed in the absence of sputtering (0 s, red line). However, in each case, there was a clear shift to higher BE values in the spectra of the metallized particles in comparison with the values associated with the bulk elements (Appendix A; Figure 4; Appendix A). In the sample after 20 s Ar^+^ sputtering (orange line), two obvious peaks with close to the theoretical ratio of 4:3 were seen for each element.

For the Au 4f spectrum the BEs of Au 4f_7/2_ were shifted from 84.0 to 84.2 eV and Au 4f_5/2_ from 88.0 to 88.2 eV. Likewise, the Pt 4f spectra were shifted from 71.2 to 72.0 eV (Pt 4f_7/2_) and from 74.0 to 75.0 eV (Pt 4f_5/2_), and the Pd 3d spectra from 335.0 to 336.2 eV (Pd 3d_5/2_) and from 340.0 to 341.8 eV (Pd 3d_3/2_).

The positions of the BE peaks of an element depend upon the oxidation state and local chemical environment. The 0.8 to 1.8 eV higher BE for the peaks associated with Pt 4f and Pd 3d, in contrast to the 0.2 eV increase of the Au 4f BE, suggests the existence of charge transfer between electron-rich Au (electronegativity value 2.54) and Pt (2.28) and Pd (2.20). This is consistent with previous reports [40,41,42] of charge transfer for bimetallic AuPd alloy formation. Likewise, in the absence of Au, ToMV particles metallized with Pt gave a lower BE than observed here [37,38]. The addition of a more electronegative element (Au) results in a decrease in the electron density around the other elements (Pt and Pd) and a concomitant increase in BE. Thus, the XPS data are consistent with alloy formation in the trimetallic ToMV particles.

The XPS spectra of all three metals in tri-metallized ToMV particles could be further deconvolved into separate small peaks, corresponding to their respective M^0^, M^+^ and M^2+^ states by curve-fitting using the carbon C 1s peak (BE 284.1 eV as a reference; Appendix A). As a control, non-metallized ToMV samples were analyzed in parallel. In tri-metallized ToMV, gold appeared to be entirely in the metallic state (Au^0^) without showing any oxide formation. The other elements, Pt and Pd, while showing traces of oxide formation, also existed mainly in the M^0^ state. The presence of all three metals predominantly in the M^0^ state suggests that the tri-metallized particles are suitable to act as electrocatalysts [43].

### 3.2. Synthesis of ToMV Bimetallic NWs (ToMV AuPt, AuPd and PtPd NWs)

In the case of bimetallic alloys, synthesis was achieved via the same metallization reaction method used for trimetallic alloys. The various two-metal (Au, Pt and Pd) combinations resulted in the fabrication of 4 nm diameter NWs within the ToMV particles and NPs on the outer surface. The bimetallic ToMV particles were characterized via TEM, HRTEM, EDX and SAED. In each case, the AuPt, AuPd and PtPd alloys showed *fcc* crystalline morphology, with lattice planes (111) and (200), and (220) and (311), as found with the trimetallic alloy (data not shown).

The elemental composition was analyzed via HAABF-STEM EDX line mapping (Figure 5) to determine whether the metal content of the bimetallic NWs was consistent along their length. The resulting EDX line profile confirmed that each bimetallic NW contained the two metals used for its fabrication. However, the relative abundance of each metal varied along the length of each NW (Figure 5). The AuPt NWs contained 73.35% Au and 27.65% Pt (Figure 5a), while the AuPd and PtPd NWs each contained 76.94% Pd and 75.60% Pd, respectively (Figure 5b,c). Thus, the approach adopted provides an effective method for fabricating bimetallic, 4 nm diameter, alloyed NWs of various compositions.

Overall, the straightforward strategy reported here can be used to produce metallic alloy nanomaterials, including trimetallic AuPtPd and bimetallic AuPt, AuPd and PtPd NWs, and for the single-step synthesis of nanoscale bi- and tri-metal alloys containing Pt, Au and Pd. This is of considerable interest since these have advantages compared to monometallic materials in electrochemical reactions due to the different reduction kinetics of Au, Pt and Pd ions [44] and their thermodynamic stability [45,46].

## 4. Conclusions

The three-metal (AuPtPd) and two-metal combination (AuPt, AuPd and PtPd) alloyed NWs were precisely synthesized within the 4 nm diameter central channel of ToMV. Au, Pt and Pd metal salt precursors were subjected to simple, wet-chemical reduction at room temperature. Ascorbic acid reduction led to the precise growth of alloyed NWs inside the ToMV rods. The presence of a polymeric capping agent, PVP_K30_ NWs, assisted in the selective NW growth within the ToMV particles. TEM, HRTEM and STEM showed that the synthesized alloyed NWs were mainly crystalline. Elemental mapping, diffraction patterns and X-ray photoelectron spectroscopy confirmed the chemical state of the metal, which is metallic M^0^.

## Figures and Tables

**Figure 1 nanomaterials-13-02705-f001:**
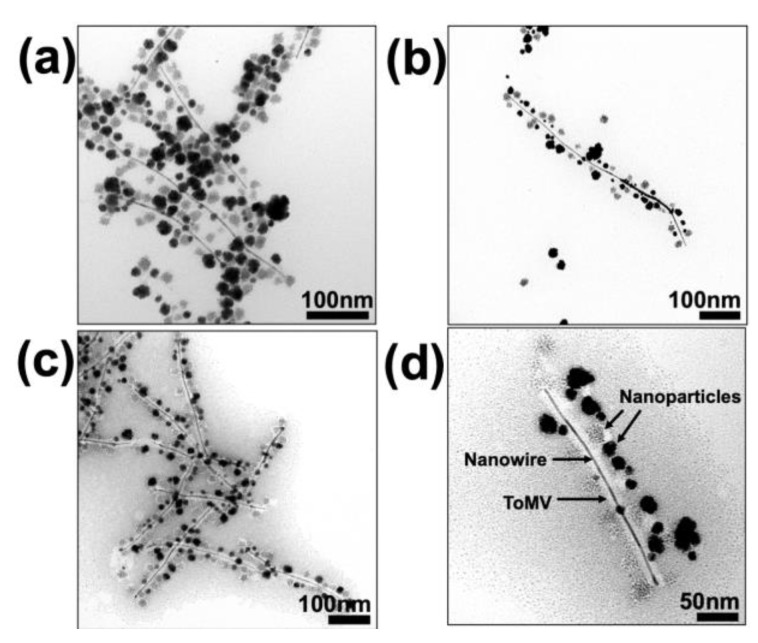
**TEM images of tri-metalized samples of ToMV particles.** Panels (**a**,**b**) are unstained images showing NW growth (dark line) and spherical NP formation on the particle surface; panels (**c**,**d**) are images collected after staining with 2% (*w/v*) aurothioglucose, which stains only the particle’s outer surface, not the inner channel of ToMV, confirming that NWs formed in the inner channel of the ToMV particles. Scale bars are shown in each panel.

**Figure 2 nanomaterials-13-02705-f002:**
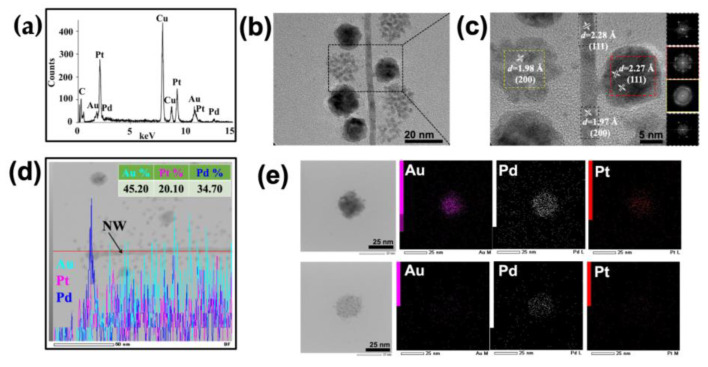
**Composition of tri-metalized ToMV particles.** (**a**) Transmission electron microscopy–energy-dispersive X-ray spectroscopy (TEM-EDX) shows the presence of all three elements, Au, Pt and Pd, in the area analyzed. Unstained TEM (**b**) and HRTEM (**c**) images show a ToMV AuPtPd crystalline NW 4 nm in diameter, and crystalline spherical NPs of diameters from 7 to 15 nm. Scanning transmission electron microscopy bright-field (STEM-BF) EDX elemental color mapping of a single AuPtPd NW (**d**), showing the distribution of the 3 metals throughout the NW, and (**e**) the presence of the 3 metals, Pt (red), Pd (white) and Au (magenta), in the NPs. A barium specimen holder and carbon-coated copper (C-Cu) EM grid were used.

**Figure 3 nanomaterials-13-02705-f003:**
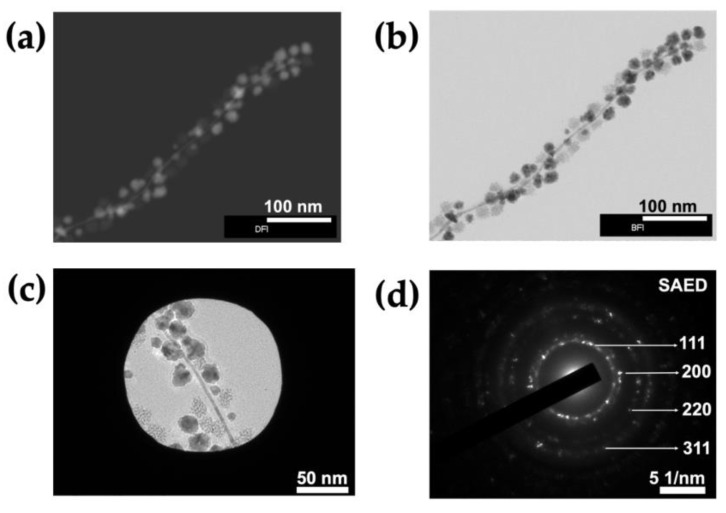
**Analysis of crystallinity of alloys in NWs and NPs.** Unstained samples were imaged using (**a**) HAADF-STEM and (**b**) HAABF-STEM. (**c**) SAED of a single trimetallic ToMV particle and stimulated diffraction ring pattern (**d**).

**Figure 4 nanomaterials-13-02705-f004:**
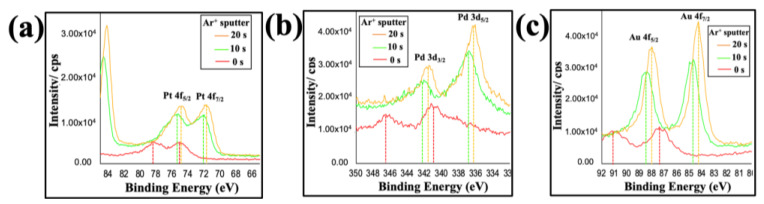
**XPS spectra of metalized ToMV particles**. Spectra corresponding to (**a**) Pt 4f, (**b**) Pd 3d, and (**c**) Au 4f d from the same sample. The effect of argon (Ar^+^) sputtered for 10 s (green) and 20 s (orange) compared to the non-sputtered argon (red) from the same sample is shown.

**Figure 5 nanomaterials-13-02705-f005:**
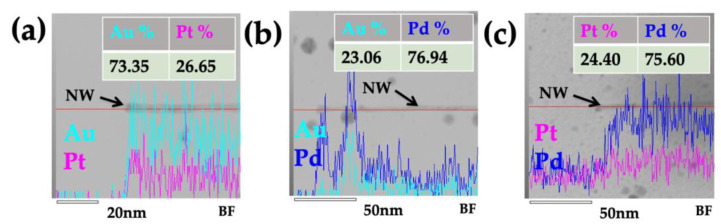
**Scanning transmission electron microscopy bright-field (STEM-BF) and EDX elemental line mapping profiles of the single bimetallic ToMV NWs (black arrow).** (**a**) ToMV AuPt, (**b**) ToMV AuPd, and (**c**) ToMV PtPd. Metal element line profile for gold (Au, cyan), platinum (Pt, magenta) and palladium (Pd, blue). A barium specimen holder and carbon-coated copper (C-Cu) EM grid were used.

## Data Availability

The research data presented in this study are available on reasonable request from the corresponding author.

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
