# Peer review of "Production of Metallic Alloy Nanowires and Particles Templated Using Tomato Mosaic Virus (ToMV)"

_nanomaterials, 2023, doi:10.3390/nano13192705_

Round 1

Reviewer 1 Report

The manuscript by Shah et al. reports a method of synthesizing metallic alloy nanowires and nanoparticles using tomato mosaic virus (ToMV) particles as templates. The authors used different combinations of gold, platinum and palladium salts to deposit alloys of various compositions on the inner and outer surfaces of ToMV particles. The resulting nanomaterials were characterized by electron microscopy, spectroscopy and diffraction techniques, which revealed their structural and chemical properties. The paper claims that ToMV particles can be used to produce uniform and crystalline alloy nanowires and nanoparticles with tunable optical, catalytic and photocatalytic properties. The claims are mostly supported, the conclusions appear to be mostly sound. My major criticism is the lack of control experiments. The authors should at least perform several parallel synthetic experiments where the ToMV is not present. Overall, I suggest a revision, and here’s a list of improvements in addition to the control experiments that the authors should consider:

  1. In the end of first paragraph, the authors claim that “it is challenging to produce AuPtPd, AuPt and AuPd alloyed nanoparticles and nanowires of uniform dimensions using purely chemical approaches”. To my knowledge, methods such as under potential deposition have been widely used to synthesize these alloyed nanoparticles with uniform dimensions. The authors should do a more thorough literature review.
  2. The authors should make their acronyms consistent. For example, nanowires and NWs were mixed used in multiple places. 

The authors should carefully check their acronyms.

Reviewer 2 Report

The manuscript entitled Production of metallic alloy nanowires and particles templated by tomato mosaic virus (ToMV) reports the detailed and exhaustive synthesis method and characterization of the bi- and trimetallic nanoparticles prepared using template technique.

The authors conducted very relevant research. The method proposed allows to synthesize nanoparticles of controlled morphology and chemical composition. The authors have conducted a thorough study of the crystalline structure and elemental composition of the nanoparticles prepared.

The authors discovered a dramatic effect - electron transfer between the constituting atoms of the alloy particle. This result makes possible to regulate the activity and selectivity of catalysts based on the synthesized nanoparticles, which will allow their use in a wide range of applications. I believe the article would have benefited a lot if the authors had given an example of the application of the original catalyst.

I have enjoyed reading this manuscript and I find no flaws in it. I think the article published will be interesting for the readers of the Journal and can be published in its current form.

Round 2

Reviewer 1 Report

The authors have sufficiently addressed my comments, I recommend acceptance in present form.